# Modelling membrane reshaping by staged polymerization of ESCRT-III filaments

Xiuyun Jiang[1,2¤a], Lena Harker-Kirschneck[1,2], Christian Vanhille-Campos[1,2,3], Anna-Katharina Pfitzner[4¤b], Elene Lominadze[1], Aurélien Roux[4], Buzz Baum[5], Anđela Šarić[1,2,3]*

**1** Department of Physics and Astronomy, Institute for the Physics of Living Systems, University College London, London, United Kingdom, **2** Medical Research Council Laboratory for Molecular Cell Biology, University College London, London, United Kingdom, **3** Institute of Science and Technology Austria, Klosterneuburg, Austria, **4** Department of Biochemistry, University of Geneva, Geneva, Switzerland, **5** Medical Research Council Laboratory of Molecular Biology, Cambridge, United Kingdom

¤a Current address: Laboratory of Soft Matter Physics, The Institute of Physics, Chinese Academy of Sciences, Beijing, China
¤b Current address: Department of Cell Biology, Harvard Medical School, Boston, Massachusetts, United States of America
* andela.saric@ist.ac.at

**Data Availability Statement:** The data and input script are available at https://github.com/sharonJXY/3-filament-model and at University

## Abstract

ESCRT-III filaments are composite cytoskeletal polymers that can constrict and cut cell membranes from the inside of the membrane neck. Membrane-bound ESCRT-III filaments undergo a series of dramatic composition and geometry changes in the presence of an ATP-consuming Vps4 enzyme, which causes stepwise changes in the membrane morphology. We set out to understand the physical mechanisms involved in translating the changes in ESCRT-III polymer composition into membrane deformation. We have built a coarse-grained model in which ESCRT-III polymers of different geometries and mechanical properties are allowed to copolymerise and bind to a deformable membrane. By modelling ATP-driven stepwise depolymerisation of specific polymers, we identify mechanical regimes in which changes in filament composition trigger the associated membrane transition from a flat to a buckled state, and then to a tubule state that eventually undergoes scission to release a small cargo-loaded vesicle. We then characterise how the location and kinetics of polymer loss affects the extent of membrane deformation and the efficiency of membrane neck scission. Our results identify the near-minimal mechanical conditions for the operation of shape-shifting composite polymers that sever membrane necks.

## Author summary

ESCRT-III proteins have the unique ability to cut membrane necks from within, which is needed for a vast number of cell remodelling events including the release of cargo-containing vesicles. ESCRT-III proteins exist in different forms, which can assemble into spiral and helical homopolymers of different curvatures, and they have been suggested to

College London Research Data Repository: https://doi.org/10.5522/04/20659479.

**Funding:** A.Š. received an award from European Research Council (https://erc.europa.eu, "NEPA" 802960), and an award from the Royal Society (https://royalsociety.org, UF160266). L. H.-K. received an award from the Biotechnology and Biological Sciences Research Council (https://www.ukri.org/councils/bbsrc/). E. L. received an award from the University College London (https://www.ucl.ac.uk/biophysics/news/2022/feb/applications-biop-brian-duff-and-ipls-summer-undergraduate-studentships-now-open, Brian Duff Undergraduate Summer Research Studentship). B. B. and A. Š. received an award from Volkswagen Foundation (https://www.volkswagenstiftung.de/en/foundation, Az 96727), and an award from Medical Research Council (https://www.ukri.org/councils/mrc, MC_CF1226). A. R. received an award from the Swiss National Fund for Research (https://www.snf.ch/en, 31003A_130520, 31003A_149975, and 31003A_173087) and an award from the European Research Council Consolidator (https://erc.europa.eu, 311536). The funders had no role in study design, data collection and analysis, decision to publish, or preparation of the manuscript.

**Competing interests:** The authors have declared that no competing interests exist.

polymerize and depolymerize with each other in a staged manner to deform and cut membranes.

We developed a computer model to explore the physical mechanisms behind vesicle budding driven by the staged assembly and disassembly of multiple elastic filaments. We identified rules that determine the outcomes of membrane remodelling, which depend on the relative physical features of the distinct filaments, the dynamics of their disassembly, and on the presence of cargo; thereby providing experimentally testable predictions. Our study provides new physical insights into the ESCRT-III-mediated vesicle budding process, at a single subunit level, and identifies the general design principles of nanomachines built from shapeshifting copolymers, which might also be realized in synthetic systems.

## Introduction

Efficient and robust remodelling of plasma and internal membranes underpins the dynamic organization of all eukaryotic cells. The ESCRT-III complex (the endosomal sorting complex required for transport III) has been identified as playing a key role in deforming and cutting membranes in a large number of cellular membrane remodelling events [1]. Distinct from other membrane-remodelling assemblies, such as clathrin, COPII and dynamin, ESCRT-III has the unique ability to constrict and cut a membrane neck from within the neck. Cellular processes that require this topological transition include the formation of multivesicular bodies within endosomes by vesicle budding, scission of the cytokinetic bridge, exosome formation, virus release, nuclear sealing and membrane repair [2–7].

Central to these activities is the fact that ESCRT-III proteins naturally occur in several isoforms of different geometries that can copolymerize into semiflexible rings, spirals, helices and cones of different lengths, widths and pitches [8, 9]. As a result, when ESCRT-III copolymers of different geometries associate with flat membranes they are able to remodel the membrane into domes, vesicles, tubules, and to catalyze the scission of membrane tubes from the inside [10–14]. Structural information of individual isoforms and some of their (co)assemblies have been revealed by recent *in vitro* experiments [15–21], and the physics on how a single ESCRT-III-like polymer can remodel the membrane has been studied in several theoretical models [22–25]. However, the physical mechanism by which different ESCRT-III copolymers, composed of various isoforms that lead to varying geometrical and mechanical properties, work together to perform rich membrane remodelling functions has not been explored.

Unlike other cytoskeletal proteins such as actin, tubulin and septins, ESCRT-III monomers do not do work by binding and hydrolysing energy-rich molecules such as ATP (although GTP binding has been revealed recently in cyanobacterial ESCRT-III homologue Vipp1 [26]). Instead, energy is provided to the system through the action of a separate ATP-consuming enzyme, the Vps4 AAA ATPase, which associates with and disassembles ESCRT-III polymers [27, 28]. This coupling of filament disassembly to the action of the Vps4 ATPase enables force production, likely by driving the polymer through a sequence of global changes in composition and geometry. Building on the pioneering work of many earlier studies [1, 29, 30], recent *in vitro* experiments showed that yeast ESCRT-III proteins exhibit striking staged co-assembly and disassembly dynamics when coupled with Vps4, which leads to membrane deformation [17, 31, 32] (see Fig 1A). These findings suggest that changes in the composition of ESCRT-III heteropolymers, driven by the Vps4 ATPase, alter the overall filament geometry to remodel the membrane. This staged filament assembly process first generates buckles in a flat

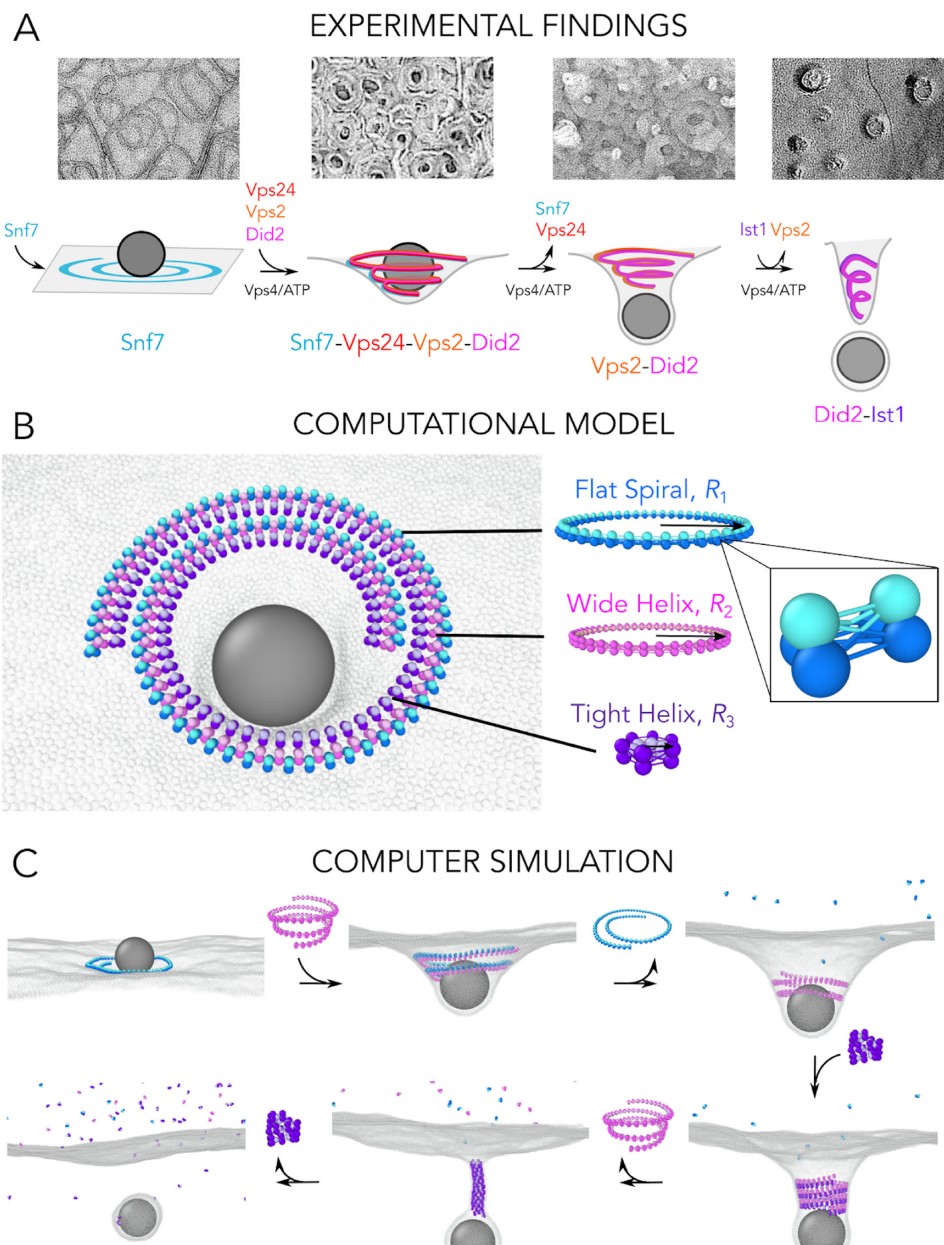

**Fig 1. Experimental findings and computational model.** A: Negative stain EM images of Snf7, Snf7-Vps24-Vps2-Did2, Vps2-Did2 and cryofreeze-fracture image of Did2-Ist1 on large giant unilamellar vesicles. Experimental protocols are detailed in S1 Appendix. The schematic cartoon representation of the ESCRT-III mediated cargo budding pathway is shown below, based on Ref. [17]. B: The model for the ESCRT-III system consists of three copolymerising filaments: a Flat Spiral (depicted in blue) of a preferential radius of curvature $R_1$, a Wide Helix (depicted in pink) of a preferential radius $R_2$, and a Tight Helix (depicted in purple) of a preferential radius $R_3$, where $R_1 > R_2 > R_3$. Each filament consists of three-beaded subunits connected by 9 bonds, which control the filament target geometry (inset). The membrane-binding face of each filament is highlighted in darker colors, and the membrane is shown in grey. A generic cargo particle corralled by the ESCRT-III filaments is shown in dark grey. C: Snapshots highlighting the stepwise activation and disassembly protocol of the three filaments, leading to fission of a cargo-loaded vesicle.

membrane sheet, then induces the formation of membrane tubules, followed by tubule constriction and fission to release cargo-containing vesicles.

Here we set out to better understand the physical mechanisms and mechanical conditions by which the sequential ESCRT-III assembly and disassembly can reshape membranes in the context of vesicle formation. To this end, we have built a coarse-grained polymer model that takes into account the membrane-bound copolymerisation of different types of ESCRT-III polymers, which differ in their geometries and mechanical rigidities [17].

Using the experimental data as a guide [17], we have modelled the ATPase-driven stepwise change in filament composition by activating filaments of different properties at different stages in the simulation. In this way, we are able to trigger transitions in the shape of the associated membrane from a flat state to a buckled state, and then to form a tubule that eventually undergoes scission. As expected, to achieve scission, copolymerising filaments need to possess different orientations of the membrane-binding interface [24], along with a stepwise increase in polymer curvatures. However, we also find that for reliable membrane-polymer association, membrane deformation and scission, it is important that the mechanical and geometrical properties of the copolymerising filaments are not too dissimilar. When looking at the dynamics, we find that optimum fission is achieved by slow changes in the composition of the polymers, and that the presence of cargo is required for scission.

## Methods and model

### Model for sequential recruitment of ESCRT-III polymers

A specific sequence of stepwise changes in ESCRT-III composition that in turn drives changes in membrane morphology has been characterised experimentally. In these vesicle budding experiments, the first ESCRT-III isoform that binds to the membrane (Snf7 in yeast, equivalent to CHMP4 in mammals) forms flat spiral polymers, without inducing any large-scale membrane deformation (Fig 1A) [33, 34]. The flat spiral polymers then recruit the second class of ESCRT-III isoforms that form helices with various radii of curvature, such as Vps24 [35] and Vps2 [18] (counterparts of the mammalian proteins CHMP3 and CHMP2, respectively). As new ESCRT-III subunits are recruited to the composite, the Vps4 ATPase drives the disassembly of the flat spiral ESCRT-III subunits. Further ESCRT-III subunits, like Did2 [36] (counterpart of mammalian protein CHMP1) and Ist1 [37] (in *in vitro* experiments at least) bind to the filament and induce a further constriction of the membrane, while the earlier subunits (Vps2 and Vps24) are disassembled. The final disassembly of Did2 and Ist1 is thought to be the last stage of the pathway in this system. Interestingly, several of these subunits are always found together on ESCRT-III composites: Vps24 is always found to co-polymerize with Vps2 on the membrane [1, 12, 17, 18, 30, 31] and Ist1 does not polymerize directly onto the membrane but instead forms a second strand on top of the Did2 strand [20, 36]. These findings suggest that, rather than a sequence of five distinct states, each corresponding to a single subunit, ESCRT-III goes through three stepwise composition changes during vesicle budding, where each state results from several subunits forming a composite.

Based on these experimental findings, to capture the general ability of these polymers to deform and cut membranes, we only consider three discrete pre-assembled filaments, each corresponding to a distinct state of the ESCRT-III composite polymer remodelling pathway (see Fig 1B), instead of the five proteins used in *in vitro* studies [17]. The first filament in the series is modelled as a Flat Spiral, which has a target radius $R_1$ and its membrane binding site is facing downwards. The second polymer is modelled as a Wide Helix with target radius $R_2$ ($R_2$ is smaller than, but close to $R_1$). The third polymer is modelled as a Tight Helix with a small target radius $R_3$ ($R_3 < R_2$), like that seen for Did2-Ist1 copolymers. The helices have their

membrane binding sites facing outwards. To replicate the experimentally observed sequence, initially only the Flat Spiral is present on the membrane, after which the Wide Helix is recruited and allowed to equilibrate. At this point we measure the membrane deformation before disassembling the Flat Spiral by severing its internal bonds. Finally, the Tight Helix is recruited, after which the Wide Helix is disassembled. Since the final filament subunits in the experiment—Ist1 and Did2—are not exchanged abruptly with the previous subunits, but rather continuously [17], we start with the Tight Helix at a radius $R_2$ and continuously constrict it to $R_3$, before disassembling the filament.

The filaments are modelled as strings of 3-beaded units connected by harmonic springs, see Fig 1B inset, as previously detailed [24]. The membrane is modelled with the one-bead-thick model [38], whose parameters are chosen to give a fluid membrane with rigidity in the physiological range of $\sim 20k_BT$. Membrane beads interact with the binding site (two darker beads) of the 3-beaded filament units via a short-ranged Lennard-Jones (LJ) attraction. We controlled the geometry of each filament—its target radius and the tilt of the membrane-binding interface—by adjusting the rest lengths of the bonds between consecutive filament subunits [24]. The individual filaments are laterally attracted to each other via a weak LJ potential between their membrane attraction sites. The system is evolved using molecular dynamics (MD) simulations within the isothermal-isobaric ensemble with the Langevin thermostat, which models the solvent implicitly. Please refer to S2 Appendix for detailed simulation protocols and a discussion on the choice of parameters.

## Results

As illustrated in Fig 1C and S1 Video, the three filaments work together as a minimal machine to remodel the membrane. This model can capture the entire process of membrane deformation and scission in a limited series of steps: i) the flat to buckle state transition, via the recruitment of the Wide Helix; ii) the buckle to tubule state transition via the subsequent disassembly of the Flat Spiral from the composite polymer; iii) the tubule constriction via replacement of the Wide Helix with the Tight Helix; and iv) the membrane scission and cargo release induced by constriction and disassembly of the Tight Helix. The same simulation protocol is also repeated with a more rigid membrane ($\sim 40k_BT$), and the same sequence of membrane remodelling events is observed (see S2 Video).

### Rigidities of the Spiral and Wide Helix should be comparable and large

It is expected that the presented sequence of polymer assembly and disassembly can exert mechanical forces and deform the membrane only under certain regimes of polymer rigidities. To explore the mechanical conditions required for membrane deformation, we first focus on the buckle deformation induced by the co-assembly of the Flat Spiral and the Wide Helix. The two filaments are equilibrated together on the membrane and the membrane deformation is measured as a function of bond stiffness for each of the two polymers (see S2 Appendix for details). As shown in Fig 2A, the competition between the rigidities of the two filaments dramatically influences the shape and the depth of the deformation. The stiffer the Spiral is, the less pronounced the deformation is. Conversely, the stiffer the Helix, the deeper the membrane deformation. The only parameter regime that allows the membrane to undergo a transition from a flat to a long-lived buckle shape, and then to a tubule upon disassembly of the Spiral, which is the scenario that has been observed in experiment [17], is the regime in which both filaments have a high and comparable stiffness (circle in Fig 2A and 2B). In this regime, the tension of the Helix provides enough energy to deform the membrane (see S1 Fig), while the tension of the Spiral provides a counterbalance to allow for the formation of a stable conical

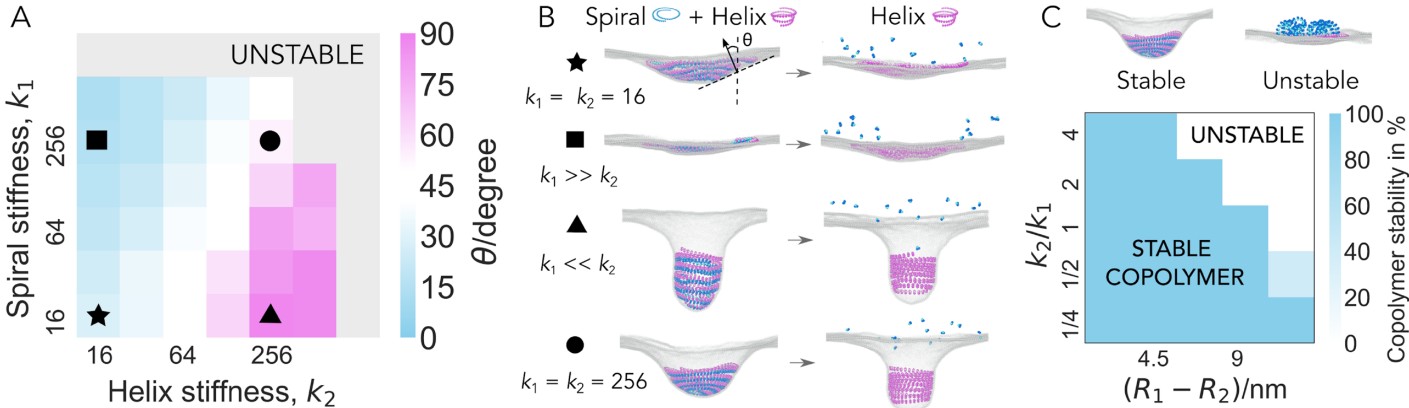

**Fig 2. Relative filament stiffness controls the membrane deformation.** A: Membrane deformation as a function of the stiffness of the Flat Spiral ($k_1$) and the Wide Helix ($k_2$), ($k_1$, $k_2$ are in units of $k_B T/\sigma^2$, $\sigma$ is the MD unit of length, $\sigma = 2.3$ nm). The membrane deformation angle $\theta$ is characterized by the angle between the vertical axis (normal to the original flat membrane plane) and the membrane normal at half depth of the deformation, as illustrated in the first snapshot in panel B. $\theta = 0°$ when the membrane is flat and $\theta = 90°$ when the membrane forms a tubule. Conical membrane deformations lie in between. The membrane deformation angle is calculated from the average of five independent runs. B: Representative simulation snapshots before and after the disassembly of the Flat Spiral, obtained for representative parameter sets as indicated by the matching labels in A. C: Copolymer stability as a function of the mismatch in the radius and ratio of stiffness between the Wide Helix and the Flat Spiral. Copolymer stability is quantified by a proportion of simulations in which the copolymerizing filaments remain attached to the membrane. The Flat Spiral is fixed at target radius $R_1 = 19.4$ nm with filament bond stiffness $k_1 = 128$ $k_B T/\sigma^2$. All data points are averaged over ten simulations.

buckle of intermediate depth. By contrast, the filament-membrane binding affinity only has a minor impact on the buckle depth, as shown in S2 Fig.

## Filament geometries cannot be too dissimilar

Spontaneous membrane scission is predicted to only occur when two bilayers come within a critical distance of one another [39, 40]. In the context of helical ESCRT-III filaments, scission has been observed when the filaments' diameter is $\sim$ 10 nm, as reported *in vitro* [17] and *in vivo* in scission of intraluminal vesicles [41] and during virus budding [42]. Generating a neck this narrow from an initially flat membrane, requires the late ESCRT-III species to be able to induce much greater curvatures than the early ones. However, a large difference between the curvatures of the copolymerizing filaments creates large geometric frustration and can easily destabilize the copolymer. Here, we explore how the relative curvature and the relative stiffness of copolymerising filaments influence the stability of the copolymer.

We keep the target radius of one filament constant, while attempting to equilibrate the other filament to a different target radius following instantaneous curvature change, and examine the copolymer stability—the probability for both of the copolymerizing filaments to remain attached to the membrane.

We find that, under conditions in which copolymerising filaments have radii that are very different from one another, membrane remodelling fails, because one of the two filaments detaches, as the other polymer induces the membrane to assume its preferred curvature (Fig 2C and S3 Fig).

Our simulations, thus far, use instantaneous assembly and sudden constriction to drive membrane deformation, which can cause filament detachment. However, one might expect different ESCRT-III polymers to be able to work together by deforming the membrane through continuous changes in copolymer structure. This could explain the existence of a large number of ESCRT-III subunit species that are involved in membrane deformation in many cellular processes.

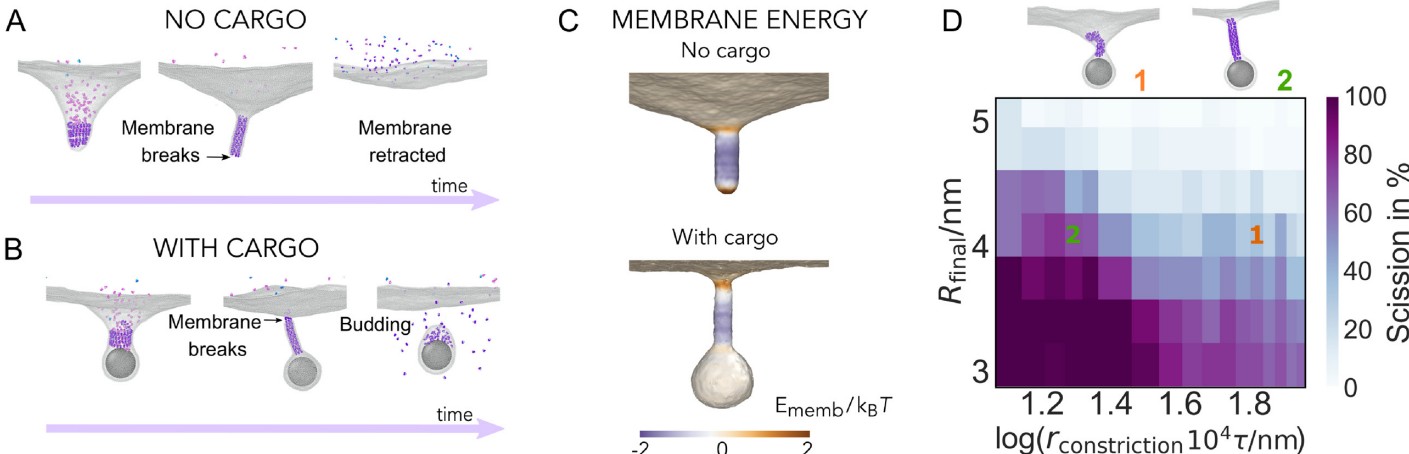

**Fig 3. Membrane scission.** The filament goes through a series of composition and geometry changes, from the Flat Spiral to the Wide Helix, and then to a constricting Tight Helix. A: In the absence of cargo, the membrane breaks at the bottom tip of the neck and retracts after the Tight Helix is disassembled. B: In the presence of the cargo, the membrane breaks at the top rim of the neck, which leads to membrane scission after the Tight Helix is disassembled. C: The local potential energy of the membrane (energy per bead) computed on representative snapshots before membrane breakage without (top) and in (bottom) the presence of the generic cargo. This energy is the sum of the membrane beads pair interaction, associated with bending and stretching, and the filament-membrane and cargo-membrane adhesion energies, relative to the flat membrane, and binned along the neck of the tube with bin width $\sigma$ and averaged over 10 snapshots. D: Scission efficiency as a function of the final target radius of the Tight Helix, $R_{final}$, and the rate of the Tight Helix constriction, $r_{constriction}$. For fast constriction, the filament partially detaches from the membrane (snapshot **1**), while for slow constriction, the filament remains attached (snapshot **2**). Each point is an average over five independent measurements.

## Membrane scission is sensitive to constriction kinetics

Data from experimental observations suggest that Ist1-Did2, the last ESCRT-III subunits to be recruited to the copolymer, are not exchanged abruptly, like the earlier polymers. This enables them to constrict the membrane to a radius that is within the fission limit [17, 43]. In our minimal model, this stage is driven by a Tight Helix that continuously constricts from the radius of a Wide Helix, $R_2$, to $R_3$, at a rate $r_{constriction}$, to mimic continuous exchange of the final subunits of the polymer. The progressive constriction is then followed by instantaneous disassembly once the filament reaches $R_{final}$.

To explore membrane neck scission, in a first instance we leave out the cargo and simply simulate the full sequence of filament exchange on the membrane (see S4 Fig and S3 Video). When a Wide Helix binds alongside the Flat Spiral, as before, we observe the formation of a membrane buckle. The buckle then turns into a tubule once the Flat Spiral is disassembled. The tubule constricts when the Tight Helix and its constriction are activated, and the tube constricts even further once the Wide Helix is disassembled. At this point the formation of membrane pores at the bottom tip of the tight tube can be observed (Fig 3A), where the curvature and curvature gradient are highest (Fig 3C). However, once the Tight Helix is disassembled within the tight neck, the neck retracts back to the mother membrane and the pores seal (Fig 3A).

The fact that the filament assembly and disassembly alone are not sufficient to cut the membrane indicates, as also previously suggested in the literature [17, 24], that the presence of cargo likely plays an important role in membrane neck scission. Indeed, when we include cargo in the same simulations, and repeat the filament exchange sequence, the membrane neck naturally severs at the top of the tube after disassembly of the Tight Helix (Fig 3B). Moreover, this process proves to be robust with respect to different cargo sizes (see S5 Fig).

Interestingly, in our simulations the scission always occurs at the top rim of the neck, closer to the mother membrane and further from the cargo (see Fig 3B). This prevents the cargo

from leaking out, even when the cargo is modelled as a collection of small particles instead of a single large particle (see S6 Fig). The membrane breakage occurs where membrane energy is the highest (see Fig 3C), and the local gradient in membrane curvature the steepest. The energy per unit surface area associated with the pore formation is similar for the simulations with and without the cargo and lies between 0.5–0.6 $k_B T/\text{nm}^2$. Decomposition of the total potential energy into the membrane mechanical energy (caused by membrane bending and stretching), filament-membrane adhesion, and cargo-membrane adhesion shows that the high potential energy at the top rim of the neck is primarily caused by the highest deformation energy due to membrane bending, with the filament and the cargo further stabilizing the membrane neck and the budding vesicle (see S7A Fig). However, we find that membrane stabilization through cargo-membrane attraction is not needed for the scission step itself, as long as the cargo is sufficiently large to create a curvature change at the rim of the neck (S8 Fig), indicating that cargo plays an important role in mediating scission by curvature generation, something that may be aided in cells by ESCRT-I and ESCRT-II complexes.

Looking more closely at the location of the scission event, we again observe the generation of small openings in the membrane. They reform and self-seal multiple times, before expanding to span a crack across the neck of the top of the tube, which resolves through membrane scission (see S4 Video). As has been previously proposed in dynamin-mediated scission [44] and consistent with the previous analytic studies [22, 23, 40], the membrane scission observed here is thus triggered by spontaneous formation of local membrane pores that occur as a consequence of filament constriction that bends and stretches the membrane.

As for the efficiency of scission (see S3 Appendix for its definition), it expectedly increases when the radius of curvature is reduced (Fig 3D). Perhaps less intuitively, the efficiency also increases with slower rates of filament constriction (Fig 3D), as fast constriction often leads to partial detachment of the filament from the membrane. This detachment can also cause the helical scaffold to coil up and become entangled within itself (see snapshot **1**, Fig 3D and S5 Video), which effectively increases the width of the filament and reduces its ability to effectively constrict the membrane. By contrast, slow (adiabatic-like) constriction of the filament allows it to maintain an ordered helical scaffold to generate a thin neck prior to scission (see snapshot **2**, Fig 3D and S6 Video). Therefore, for the same amount of constriction, the majority of simulations fail under fast constriction, but are successful when constriction is slow.

For dynamin-mediated scission, it has been reported that shorter polymers appear to be more fission competent [45–47]. We explore whether the polymer length has a similar effect in ESCRT-mediated fission. To this end, we repeat the Tight Helix scission simulations with various filament lengths. Similarly to the dynamin case, we find that scission efficiency increases substantially for shorter filaments (S9 Fig), with the slower constriction still leading to higher scission efficiency.

## Membrane scission is sensitive to disassembly kinetics

Studies of dynamin- [45, 48] and ESCRT-III- [49, 50] mediated membrane scission have highlighted the importance of filament disassembly. In the case of ESCRT-III filaments, Vps4 is needed for the change in filament composition through its ability to regulate the timing of the disassembly of each ESCRT-III polymer. This prompts us to use simulations to test the impact of different disassembly protocols of the final polymer on membrane scission.

We first examine the role of subunit disassembly order (Fig 4A). In the instantaneous disassembly protocol, all subunits within the final polymer are deactivated at once. In the random disassembly protocol, we let the subunits of the filament disassemble in a random fashion to mimic the stochastic association of the Vps4 disassembly factor with its target polymer. We

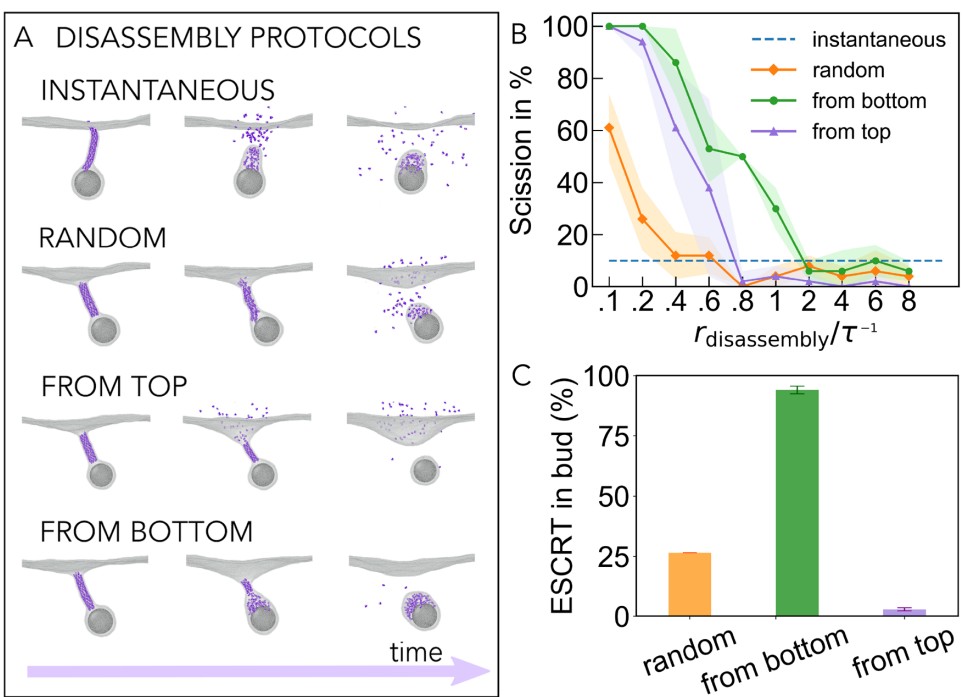

**Fig 4. The role of final helix disassembly in neck scission.** A: Four different protocols for the Tight Helix disassembly are explored, as shown with the representative snapshots along trajectories: instantaneous, randomised, sequential from the top, and sequential from the bottom. B: Membrane scission efficiency as a function of rate of filament disassembly for the different disassembly protocols ($R_{\text{final}}$ = 5.7 nm). All the data is collected for $r_{\text{constriction}}$ = $2.1 \times 10^{-3}$ nm/$\tau$, averaged over 5 independent measurements, and the standard deviation is indicated by the shaded area. C: Fraction of filament monomers located in the budded vesicle after successful scission.

also implement a sequential filament disassembly protocol initiated from one end—the top or bottom of the filament—and propagated to the other end. This can serve to model a situation in which Vps4 can only access monomers at the termini of crowded polymers in membrane tubes. This is relevant because Vps4 complexes are tens of nanometers in diameter, which may prevent the complex from entering the tight membrane neck, forcing it to act from one end. Interestingly, in every case, as long as disassembly is not instantaneous, the protocols are able to induce neck scission relatively well for the chosen target radius of the Tight Helix (see Fig 4B, solid versus dashed lines), when the rate of filament disassembly is slow. Furthermore, slower disassembly rates also lead to more reliable scission (see Fig 4B), presumably because the membrane has more time to relax and adapt to locally applied forces.

To our surprise, the two sequential disassembly protocols are able to achieve much higher scission efficiency than the randomised protocol for a given slow rate of filament disassembly (Fig 4B). In examining why this might be the case, we observe that the disassembly of polymer in the middle of the filament in the randomised protocol causes filament sections to be trapped inside the neck, which creates steric pressure that opposes membrane constriction. By contrast, sequential disassembly allows monomers to leave the tight neck in an ordered fashion, so that the membrane is able to relax and undergo spontaneous scission [51]. Finally, the location of the disassembled ESCRT-III monomers depends strongly on the disassembly protocol used. The majority of subunits are found in the budded vesicle if disassembly occurs sequentially from the bottom, and in the mother compartment if disassembly occurs sequentially from the top (see Fig 4C). This partitioning of subunits is found to be relatively insensitive to the rate of disassembly.

## Discussion

In this paper we describe a minimal coarse-grained model for ESCRT-III-driven membrane vesicle formation based on sequential polymer recruitment and disassembly [17]. By simplifying the pathway to consider only three distinct filaments successively dominating the copolymer, we are able to explore the generic features of this system.

Using this model we are able to show that two essential steps are required to change the resting geometry of the copolymer: i) the change of the filament's membrane binding surface from the base to the outer side of the filament (the transition from the Flat Spiral to the Wide Helix); and ii) a decrease in the radius of curvature of the filament so that it reaches a radius close to the fission limit (the transition from a Wide Helix to a Tight Helix). These findings are consistent with our previous work in which ESCRT-III-driven membrane remodelling is described through global geometry changes of a single filament [24]. Here, such global changes of the filament are explained through explicit exchanges in its copolymer composition, and the roles of the properties of different polymers and the protocols for their exchanges are explored.

For the process to work robustly and efficiently, we show that the three filaments need to have increasing, but not too dissimilar, curvatures and rigidities. This finding is in line with conclusions from experiments and numerical modelling of the mechanical properties of yeast ESCRT-III polymers [17, 19]. *In vitro* experiments show that Snf7 can bind to flat membranes [33], while the Did2-Ist1 complex binds to membrane tubes [20]. Although the exact preferred orientations of Vps24 and Vps2 on the membrane is not yet clear, their ability to induce membrane deformation when combined with Snf7 [9, 17, 18], and their ability to form helical polymers [19, 21], suggests that they have non-zero tilt or twist.

Our minimal model tries to capture the general features of the physical mechanism of membrane-deformation and cargo release rather than model a specific set of ESCRT-III polymers. However, the Flat Spiral appears to be a relatively good match for Snf7, which forms a flat spiral that initiates the process. The Wide Helix maps reasonably well onto the Vps2-Did2 copolymer, which induces membrane buckling, and the Tight Helix appears a good model for the Did2-Ist1 copolymer, which forms membrane tubes. In reality, more strands with intermediate geometries may be involved in yeast and animals (e.g., Vps24-Vps2), since as we show, this can enhance the robustness of the membrane reshaping processes. This may be one of the reasons why the number of sub-types of ESCRT-III subunit is so high in many eukaryotes [49, 52]. In some instances though, e,g. in Asgard archaea, whose genomes only code for two ESCRT-IIIs and Vps4 [53], it may be that ESCRT-I and II compensate for the loss of one.

We find that the presence of cargo is necessary for scission, as it enforces the curvature change, stabilizes the nascent bud and prevents retraction of the membrane to its mother membrane reservoir. For scission to occur, we expect the cargo does not need to be solid, as modelled in our simulations, but can also take the form of a membrane-bound dense fluid or membrane-inserted cluster, as long as it prevents membrane retraction upon filament disassembly. Slower rates of filament remodelling and disassembly increase the robustness of membrane deformation and scission, which, together with the findings on the location of membrane breakage with and without the presence of cargo, may be experimentally testable by using Vps4 mutants that change the rates of polymer exchange and disassembly and/or changes in Vps4-ESCRT-III binding, for instance. In addition, we expect there might be a trade-off between the robustness of the process and the necessity for it to reach a certain speed to fulfill its biological role.

We note that our model is based on preassembled polymers, preventing us from capturing the effects of dynamic filament assembly and copolymerisation, something that may introduce

additional layers of complexity to the system. We have recently investigated the rules for staged assembly and disassembly of different ESCRT-III isoforms on a deformable membrane using chemical kinetics and theory of elasticity [54]. The development of particle-based coarse-grained models that allow the analysis of the dynamic assembly remains an important task for the future. Beyond the ESCRT-III system, our study identifies physical principles for the existence of shape-shifting membrane-reshaping copolymers, which might be realised in synthetic systems, such as those made of DNA building blocks.

## Supporting information

**S1 Appendix. Experimental protocols.**
(PDF)

**S2 Appendix. Simulation protocols.**
(PDF)

**S3 Appendix. Analysis protocols for simulation.**
(PDF)

**S1 Table. Summary of all bead-bead interactions (A-B) and their default interaction strength.**
(PDF)

**S2 Table. Conversion table to map values from MD units to physical units.**
(PDF)

**S1 Fig. Effects of the Wide Helix rigidity on membrane deformation.**
(PDF)

**S2 Fig. Effects of membrane-binding affinity of the Spiral and the Helix on membrane deformation.**
(PDF)

**S3 Fig. The stability of the copolymer formed by Wide Helix and Tight Helix.**
(PDF)

**S4 Fig. Typical snapshots along the trajectory where the three filaments are activated and disassembled in a stepwise manner in the absence of cargo.**
(PDF)

**S5 Fig. Impact of cargo size on membrane scission.**
(PDF)

**S6 Fig. Tight Helix constriction and scission with six small, volume-excluded cargos.**
(PDF)

**S7 Fig. The local potential energy of the membrane and the energetic decompositions computed on representative snapshots before membrane breakage.**
(PDF)

**S8 Fig. Tight Helix constriction and scission when the cargo-membrane adhesion is replaced by volume-exclusion.**
(PDF)

**S9 Fig. Shortening the length of the Tight Helix promotes scission.**
(PDF)

**S1 Video. Stepwise activation and disassembly of the three filaments in the presence of the cargo.**
(MP4)

**S2 Video. Stepwise activation and disassembly of the three filaments in the presence of the cargo using membrane rigidity $\kappa \sim 40k_\mathrm{B}T$ ($\mu = 5$ in the membrane model).**
(MP4)

**S3 Video. Stepwise activation and disassembly of the three filaments in the absence of the cargo.**
(MP4)

**S4 Video. Small pores form and reseal multiple times before scission.** The video pauses for a second when pores are detected.
(MP4)

**S5 Video. Fast Tight Helix constriction: Most trajectories fail in the fission.** Rate of constriction $r_\mathrm{constriction} = 6.9 \times 10^{-3}$nm/$\tau$ is used.
(MP4)

**S6 Video. Slow Tight Helix constriction: Most trajectories achieve fission.** Rate of constriction $r_\mathrm{constriction} = 2.1 \times 10^{-3}$nm/$\tau$ is used.
(MP4)

## Acknowledgments

We thank Billie Meadowcroft and Ivan Palaia for helpful discussions. The authors acknowledge the use of the UCL Myriad High Performance Computing Facility (Myriad@UCL), and associated support services, in the completion of this work.

## Author Contributions

**Conceptualization:** Aurélien Roux, Buzz Baum, Anđela Šarić.

**Data curation:** Xiuyun Jiang, Elene Lominadze.

**Formal analysis:** Xiuyun Jiang.

**Funding acquisition:** Aurélien Roux, Buzz Baum, Anđela Šarić.

**Investigation:** Xiuyun Jiang, Anna-Katharina Pfitzner, Elene Lominadze.

**Methodology:** Xiuyun Jiang, Lena Harker-Kirschneck, Christian Vanhille-Campos.

**Project administration:** Anđela Šarić.

**Resources:** Anđela Šarić.

**Supervision:** Anđela Šarić.

**Visualization:** Xiuyun Jiang, Christian Vanhille-Campos, Anna-Katharina Pfitzner, Elene Lominadze.

**Writing – original draft:** Xiuyun Jiang, Anđela Šarić.

**Writing – review & editing:** Xiuyun Jiang, Lena Harker-Kirschneck, Christian Vanhille-Campos, Anna-Katharina Pfitzner, Aurélien Roux, Buzz Baum, Anđela Šarić.

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
