## [Decision Letter · Decision Letter 0]

18 Jun 2022

Dear Prof. Saric,

Thank you very much for submitting your manuscript "Modelling membrane reshaping by staged polymerization of ESCRT-III filaments" for consideration at PLOS Computational Biology. As with all papers reviewed by the journal, your manuscript was reviewed by members of the editorial board and by several independent reviewers. The reviewers appreciated the attention to an important topic. Based on the reviews, we are likely to accept this manuscript for publication, providing that you modify the manuscript according to the review recommendations.

Sincerely,

Nir Gov

Associate Editor

PLOS Computational Biology

Nir Ben-Tal

Deputy Editor

PLOS Computational Biology

[LINK]

Reviewer's Responses to Questions

**Comments to the Authors:**

Reviewer #1: In this article, Jiang et al. use a coarse-grained computational model to understand how sequential (co-)polymerization of different filaments on a membrane can induce membrane remodeling. In particular, they apply this model to investigate mechanisms by which ESCRT-III filaments induce membrane fission. Overall, this is a very well-written and interesting study, that presents novel computational results of a simple model for ESCRT-III filament action on membranes. I have no major comments on the correctness of the paper, as said, I think it will be a valuable addition to the field and hopefully will triggere further experimental work on in vitro scission by ESCRTS. Along these lines I only have a few comments, which are basically aimed at helping the readers:

- Based on the results of the model, could the authors suggest some experiments to validate some of the predictions of the model? I feel this could help experimental biophysicists see the relevance of the findings herein presented.

- The section named "Model for sequential recruitment..." is generally clear and well written, but I think it could help the readers to present tighter connections with the experiemnts (especially in vitro experiments). For example, in the model only 3 filaments are included. Is there something known whether 3 filaments (possibly formed by the 5 proteins mentioned by the authors) are enough to drive membrane remodelling?

- The authors use a bilayer with a bending rigidity of 15 kBT. This is fine, probably in the low range of biological bilayers. ESCRTs however act at different membranes within the cell, which could/might have different rigidities (up to ~40 kBT possibly). Have the authors tested whether the actual value of kappa (within a factor of 2) has any significant role in the observed effects?

- The fission pores observed in the simulations, are always non-leaky, or did the authors observed in some cases leaky fission?

- When the authors say (pg. 5, last paragraph) that scission always occurs where the energy is highest, is that because in the model, the interaction of the cargo with the membrane is large enough to stabilize the bottom part of the tube? What would happen if the cargo is only anchored through a point (e.g. to the tip of the invagination?).

- In the methods (pg 9, last paragraph), the authors say they use DBSCAN to measure scission efficiency. I think more details are needed here, such as how this is done (e.g. epsilon, Nmin values, etc.)

- There are other theoretical studies on how ESCRTs remodel membranes (e.g. Lenz et al. PRL 2009; Fabrikant et al. PLoS Comput Bio. 2009). If they consider it appropriate, the authors should consider discussing similarities/differences with these previous reports.

Reviewer #2: This manuscript by Jiang et al. presents a computational investigation into the function of protein complex ESCRT-III, a ubiquitous machinery used by the cell to sever its membrane from the inside. This action is known to involve many distinct proteins that successively form several membrane-associated filaments with different shapes, resulting in eventual membrane rupture. Experimental information on this process essentially consists in static, high-resolution structural data and dynamic studies providing little spatial information due to their limited optical resolution. Combined with its complexity, these factors result in substantial uncertainty as to the mode of action of the ESCRT-III complex. This implies that a detailed understanding of the physical constraints governing this process has a potential to significantly narrow down the space of possible mechanisms and thus further our understanding of this crucial biological problem.

In their study, the authors design an elegantly simplified model of the ESCRT-III assembly and deformation sequence. They are therefore able to validate a physically consistent, detailed scenario for the resulting membrane severing action for the first time while providing many valuable insights into the mechanical requirements on the protein filaments involved. Given the current state of knowledge on this problem, I believe that this study strikes an excellent balance between the inclusion of biologically relevant details such as the succession of several filament species, and the necessity to simplify the problem to extract valuable insights as well as keep the number of unknown parameters under control. They also do a great job of keeping the discussion intelligible despite the relative abundance of such parameters. I wholeheartedly recommend the publication of this manuscript, and request that the authors will implement the following recommendations which I believe will help amplify the impact and usefulness of their work.

---

Detailed comments

- In the initial description of the polymer assembly sequence, specifying from the onset that $R_2$ is smaller than but close to $R_1$ would help the reader form an accurate early picture of the process.

- The approach chosen by the authors to investigate the influence of the various mechanical parameters is to choose a certain set of parameter values, and to then vary individual parameters around these values while holding the other parameters fixed. Some parameters are not varied, such as the filament-to-membrane sticking energy. It would of course be unreasonable to expect the authors to explore the whole 7- or more-dimensional parameter space of their system. That said it would be nice if they could explain how they chose the central set of parameter values that they perturb around. Were these inspired by some measurements from the literature, or are they reasonable orders of magnitude inspired by similar systems? A discussion would be welcome, perhaps accompanied by a table of the model parameters and reasons for the choice of their values. In addition to the parameters that are already explicitly discussed, I would be interested in knowing how the initial filament length is chosen and whether the authors expect that much will depend on this parameter.

- Another related point is that the authors mostly discuss the values of their parameters in simulations units. This clearly makes for a cleaner paper overall, but providing a (putative?) dictionary between these and physical units somewhere would make it quicker and easier to relate these results to other studies, be they theoretical or experimental. One especially valuable piece of this discussion would be a better understanding of the typical time scales at which the transition pictured in Fig. 4B takes place.

- Fig. 2 & corresponding text: The authors state that an excessive value of $k_2/k_1$ or of $R_1-R_2$ leads to an unstable filament attachment. Couldn't this be compensated for by making the filament more sticky to the membrane? This is of course connected to the previous item - if a rationale were given for the value of the stickiness used in the simulations, then this wouldn't be much of a concern.

- In Fig. 3C, the color bar measures $E_\\text{memb}$ in units of $k_BT$. It would make more sense to me to describe the membrane energy as an energy per unit surface since the membrane "beads" don't have a straightforward real-life counterpart. This would mean giving this energy in units of $k_BT/\\sigma^2$ or $k_BT/nm^2$.

- p. 4 and Fig. 2A,B : the authors state that $k_1 \\simeq k_2$ large is required for deformation, but the data of Fig. 2A,B suggests that $k_1 \\ll k_2$ would also work. Could the authors clarify why they rule out this regime ? Is this because of the increased risk of a Fig. 2C-type detachment?

- The three-polymer model is nice and simple, but why not even fewer? I would be interested in having the authors quickly discuss whether they tried a "Flat spiral -> Tight helix" scenario and why it did not work.

- The "tight helix disassembly triggers fission" effect is perhaps the most interesting physical puzzle brought about by the simulations. This sounds reminiscent of a proposal for the function of dynamin by Bashkirov and co-workers (DOI: 10.1016/j.cell.2008.11.028), although I was never able to make intuitive sense of the physical mechanism at play in their model. Could the authors discuss their findings in relation to this more complicated model? This also sounds a bit reminiscent of the breaking mechanism of Ref. [29], which is also a bit unclear to me (despite my being an author of the paper!). Does the authors' result shed new light on these earlier ideas? And regardless of earlier papers, can the authors speculate on how disassembly causes breaking?

- p.6 : "Finally, the location of the disassembled ESCRT-III monomers depends entirely on the disassembly sequence used" : this feels like it depends on the value of the diffusion coefficient used in simulation units. Again, it would be helpful to be able to relate these quantities to known orders of magnitudes in physical units.

- Unless I have missed it, the authors never mention the spontaneous pitches of their two helices. This should be done somewhere. Is it zero as Fig. 1B suggests?

---

Typographical errors & clarifications

- abstract: "We set to" -> "We set out to"

- Fig. 3 caption: "for fast constriction, filament partially detaches" -> "for fast constriction, /the/ filament partially detaches"

- Fig. 3 caption: "for slow constriction, filament remains attached" -> "for fast constriction, /the/ filament remains attached"

- p.4 section title: "Rigidities of the Spiral and Helix should be comparable and large" -> "Rigidities of the Spiral and /Wide/ Helix should be comparable and large" (to avoid a possible confusion with the tight helix)

- p.4: "The wider the Helix is, the deeper the membrane deformation" -> "The wider the Helix, the deeper the membrane deformation"

- p.6: "The protocols [...] induce membrane scission relatively good" -> "relatively well" or "relatively efficiently"? "good" sounds a bit strange.

- p.6: "the two protocols [...] cut larger membrane necks than the randomized protocol" : a word is missing between "neck" and "than".

- p.6: "the Did2-Ist1 complex bind" -> either "complexes" or "binds"

- p.7: "up to the smallest size" -> "down to the smallest size"

- p.8: The axis coordinates "x" and "y" should be in italics/mathematical notation.

- "Time step of [...] is used" -> "A time step of [...] is used"

- "Lammps" -> "LAMMPS"?

- p.8: "rest of the system": do the authors mean the cargo, or is there something else that I am not seeing?

- p.8: some intra-filament bonds appear to be Lennard-Jones (filament-filament binding) and others harmonic (top right of the page). Which is which is a little unclear. I think it would be useful to have a slightly slower, clearer description of what the interactions are between their many different types of beads. Maybe include a matrix-like table?

- p.8" "tilted ring": I do not understand this term. Is this another name for a helix? Or is this the same kind of objects as in Ref. [16]? In either case the value of the tilt parameter - pitch or angle - should be given. The term "tilt" also appears in p.9 without being more understandable.

- p.9: "$R_{init}$ to nm $R_{final}$": it looks like "nm" should be removed.

- p.10: The authors of Ref. [1] appear to be missing.

-- Martin Lenz

**Have the authors made all data and (if applicable) computational code underlying the findings in their manuscript fully available?**

Reviewer #1: **No: **As far as I've seen, the codes for the computational model are not published or linked to in this manuscript.

Reviewer #2: Yes

PLOS authors have the option to publish the peer review history of their article (what does this mean?). If published, this will include your full peer review and any attached files.

Reviewer #1: **Yes: **Felix Campelo

Reviewer #2: **Yes: **Martin Lenz

Figure Files:

Data Requirements:

Reproducibility:

References:

---

## [Decision Letter · Decision Letter 1]

19 Sep 2022

Dear Prof. Saric,

We are pleased to inform you that your manuscript 'Modelling membrane reshaping by staged polymerization of ESCRT-III filaments' has been provisionally accepted for publication in PLOS Computational Biology.

Best regards,

Nir Gov

Academic Editor

PLOS Computational Biology

Nir Ben-Tal

Section Editor

PLOS Computational Biology

Reviewer's Responses to Questions

**Comments to the Authors:**

Reviewer #1: All my questions and comment have been convincingly addressed by the authors.

Reviewer #2: I am happy with the authors' responses to my and the other reviewer's comments and recommend publication of this nice study.

**Have the authors made all data and (if applicable) computational code underlying the findings in their manuscript fully available?**

Reviewer #1: Yes

Reviewer #2: Yes

PLOS authors have the option to publish the peer review history of their article (what does this mean?). If published, this will include your full peer review and any attached files.

Reviewer #1: **Yes: **Felix Campelo

Reviewer #2: **Yes: **Martin Lenz

---

## [Editor Report · Acceptance letter]

12 Oct 2022

PCOMPBIOL-D-22-00700R1 

Modelling membrane reshaping by staged polymerization of ESCRT-III filaments

Dear Dr Šarić,

I am pleased to inform you that your manuscript has been formally accepted for publication in PLOS Computational Biology. Your manuscript is now with our production department and you will be notified of the publication date in due course.

With kind regards,

Zsofia Freund
